# Nonvolatile ferroelectric field-effect transistors

Xiaojie Chai[1,6], Jun Jiang[1,6], Qinghua Zhang[2], Xu Hou [3,4], Fanqi Meng[2,5], Jie Wang [3,4], Lin Gu [2], David Wei Zhang[1] & An Quan Jiang [1✉]

Future data-intensive applications will have integrated circuit architectures combining energy-efficient transistors, high-density data storage and electro-optic sensing arrays in a single chip to perform in situ processing of captured data. The costly dense wire connections in 3D integrated circuits and in conventional packaging and chip-stacking solutions could affect data communication bandwidths, data storage densities, and optical transmission efficiency. Here we investigated all-ferroelectric nonvolatile LiNbO$_3$ transistors to function through redirection of conducting domain walls between the drain, gate and source electrodes. The transistor operates as a single-pole, double-throw digital switch with complementary on/off source and gate currents controlled using either the gate or source voltages. The conceived device exhibits high wall current density and abrupt off-and-on state switching without subthreshold swing, enabling nonvolatile memory-and-sensor-in-logic and logic-in-memory-and-sensor capabilities with superior energy efficiency, ultrafast operation/ communication speeds, and high logic/storage densities.

[1] State Key Laboratory of ASIC & Systems, School of Microelectronics, Fudan University, 200433 Shanghai, China. [2] Beijing National Laboratory for Condensed Matter Physics, Institute of Physics, Chinese Academy of Sciences, 100190 Beijing, China. [3] Department of Engineering Mechanics, Zhejiang University, 310027 Hangzhou, China. [4] Key Laboratory of Soft Machines and Smart Devices of Zhejiang Province, 310027 Hangzhou, China. [5] University of Chinese Academy of Sciences, 100049 Beijing, China. [6] These authors contributed equally: Xiaojie Chai, Jun Jiang. ✉email: aqjiang@fudan.edu.cn

The frequent data shuttling between the physically separated processing and memory units in traditional digital computers incurs considerable penalties on the energy efficiency and data bandwidth, which is further intensified by the increasing disparity between the speed of the memory unit and the processor[1]. The latest generation of computing approaches requires memory devices to enable high-throughput, energy-efficient, and area-efficient information processing. Ferroelectric materials with spontaneous polarizations have piezoelectric, pyroelectric, and electro-optic properties and are widely used in nonvolatile memories, sensors, and electro-optic modulators that rely on heterogeneous integration of field-effect transistors (MOSFETs)[2–5]. Recently, there have been many observations of ferroelectric single crystals in which the domain walls are highly conductive[6–12]. Hall voltage measurements of $ErMnO_3$ and $YbMnO_3$ consistently indicated high mobilities of ~670 cm$^2$ V$^{-1}$ s$^{-1}$ for active p-type carriers in tail-to-tail domain walls at room temperature[13,14]. These walls can be reversibly created, positioned and shaped using electric fields on a femtosecond time scale[15], which is encouraging for future high-speed domain-wall nanoelectronics[16–18]. Although control over the injection of conducting domain walls between the source and drain electrodes has allowed landmark demonstration of three-terminal resistance memory devices[19,20], they lack the superior abrupt on-and-off state switching capability of nonvolatile transistors. This functionality is pivotal for the performance of in situ data computing, storage, and sensing operations[21].

Here we demonstrate such transistors using 5 mol% MgO-doped $LiNbO_3$ (LN) monodomain single crystals that offer significantly enhanced diode-like wall currents[10,11]. LN single-crystal thin films can currently be produced by ionic slicing of surface layers off bulk crystals and chemical bonding of these layers to Si wafers, and have been widely applied in pyroelectric sensors, surface acoustic wave filters, and electro-optic modulators[3,4]. In this work, conducting drain–gate and drain–source conduits were complementarily created and erased using a controlled gate/source voltage through 180° reversal of the local domains. These junctionless transistors demonstrate low leakage, fast operating speeds (<5 ns), high wall currents (~110 μA μm$^{-1}$), and abrupt off-and-on state switching without subthreshold swings (SS). In addition, these transistors offer uniform, programmable threshold voltages like those of the nanowire Ge/Si core/shell FET, with computing, memory, and addressing capabilities[22].

## Results

### Device design and electrical measurement of LN FET

Figure 1a (Supplementary Fig. 1a–c) displays phase-field simulations of the switched domains (thick arrows) and conducting walls between the drain (D), gate (G), and source (S) electrodes for a cross-sectional three-terminal LN cell (S2) controlled via the gate voltage ($V_g$) under application of sufficiently high drain voltage ($V_d$) when the source voltage is grounded ($V_s = 0$). Because $V_g$ is smaller than the forward threshold voltage ($V_{t1}$), the electric field ($E$) between D and G is strong enough to reverse the D–G domain during head-to-head D–G wall formation, as shown in Step 1. The charged D–G wall in the thickness of 1.06 nm (inset in Supplementary Fig. 1a) is conducting and is partially screened by free electrons near G to compensate for the polarization charge ($P$) of 70 μC cm$^{-2}$ to reduce the depolarization energy[23]. At this instant, the D–G current ($I_{dg}$) is on but the D–S current ($I_{ds}$) is off. When $V_g$ increases above $V_{t1}$, $E$ strengthens between G and S, which allows the D–G domain to grow throughout the entire cell in forming a D–S wall to turn on $I_{ds}$, as illustrated in Step 2. Simultaneously, the previous conduit between D and G is blocked (i.e., $I_{dg} = 0$). When $V_g$ is reduced again below the backward threshold voltage $V_{t2}$, the D–S

**Table 1 Samples 1–13 with various geometrical sizes.**

| Sample | $l_{dg}$ (nm) | $l_g$ (nm) | $l_{gs}$ (nm) | $h$ (nm) |
|--------|------------|---------|------------|--------|
| S1  | 70  | 200 | 70  | 55 |
| S2  | 106 | 269 | 91  | 55 |
| S3  | 127 | 242 | 130 | 55 |
| S4  | 148 | 241 | 151 | 55 |
| S5  | 175 | 397 | 192 | 55 |
| S6  | 70  | 200 | 57  | 55 |
| S7  | 128 | 250 | 139 | 55 |
| S8  | 203 | 85  | 280 | 64 |
| S9  | 112 | 57  | 189 | 64 |
| S10 | 101 | 200 | 126 | 70 |
| S11 | 117 | 50  | 163 | 55 |
| S12 | 135 | 50  | 150 | 55 |
| S13 | 106 | 63  | 111 | 65 |

domain contracts back into the Step 1 state to shut $I_{ds}$ off via reversal of the local G–S domain in Step 3. This repetitive redirection of the conducting domain walls between D, G, and S appears similar to a single-pole, double-throw digital switch controlled by the gate voltage. Normally, $V_{t1} \neq V_{t2}$ because of hysteretic domain switching behavior that can be defined using the geometrical sizes of the samples (S1–13) listed in Table 1.

Figure 1b shows planar scanning electron microscope (SEM) image of S9, where $l_{dg}$ and $l_{gs}$ are the distances between D and G and between G and S in widths of $w$, respectively, and the gate electrode width is $l_g$. This mesa-like transistor was etched to a depth ($h$) of 64 nm with left and right side slopes (~80°) to contact D and S fully using Pt electrodes that were fabricated on an X-cut LN surface via electron-beam lithography (EBL) and dry etching processes (see the "Methods"). Figure 1c shows double $I_{ds}-V_d$ and $I_{dg}-V_d$ curves between 0 and 8 V when $V_g = V_s = 0$ V. During the first $V_d$ sweep from 0 to 8 V, the initially off $I_{dg}$ ($I_{ds}$) begins to turn on (off) at 2.6 V before turning off (on) again at 5.7 V. Subsequently, the off (on) state is maintained until $V_d$ drops to 0. The complementary on/off $I_{dg}$ and $I_{ds}$ follow the phase-field simulation results for wall redirection between G, D, and S in Fig. 1a (see Steps 1 and 2). During the next repeated sweep, $I_{dg}$ ($I_{ds}$) is always off (on above 2.2 V), indicating the nonvolatile natures of the D–G and D–S walls when created. Because the device was poled at an opposite voltage of −8 V, the on-and-off state switching above is reproduced in the first sweep. Subsequent in-plane piezoresponse force microscopy (PFM) phase imaging showed the remnants of the partially switched D–G domain after poling at +5 V (Step 1), which later extended throughout the D-S cell at +8 V (Step 2), as indicated by the yellow-colored regions in Fig. 1d.

When the D–S wall has been created, $I_{ds}$ can be controlled using $V_g$ through a major carrier accumulation/depletion mechanism within the unchanged D–S conduit without invoking of domain motion between D, G, and S, similar to a volatile MOSFET. Figure 1e shows semi-logarithmic $I_{ds}-V_g$ plots at different $V_d$ values for S8 with much longer $l_{dg}$ and $l_{gs}$ to avoid D–S wall redirection during sweeping within the voltage range studied. Normally, the forward and backward on/off switching curves are hysteretic (Supplementary Fig. 2a) that can be minimized by reducing the D and S electrode contact heights to the LN mesa for S13 (Supplementary Fig. 2b). From the solid-line fittings shown in Fig. 1e, we consistently estimated SS values as high as 216 mV dec$^{-1}$.

### $V_g$-controlled transistor without subthreshold swing

Progressive downscaling of the operating voltages and device sizes of

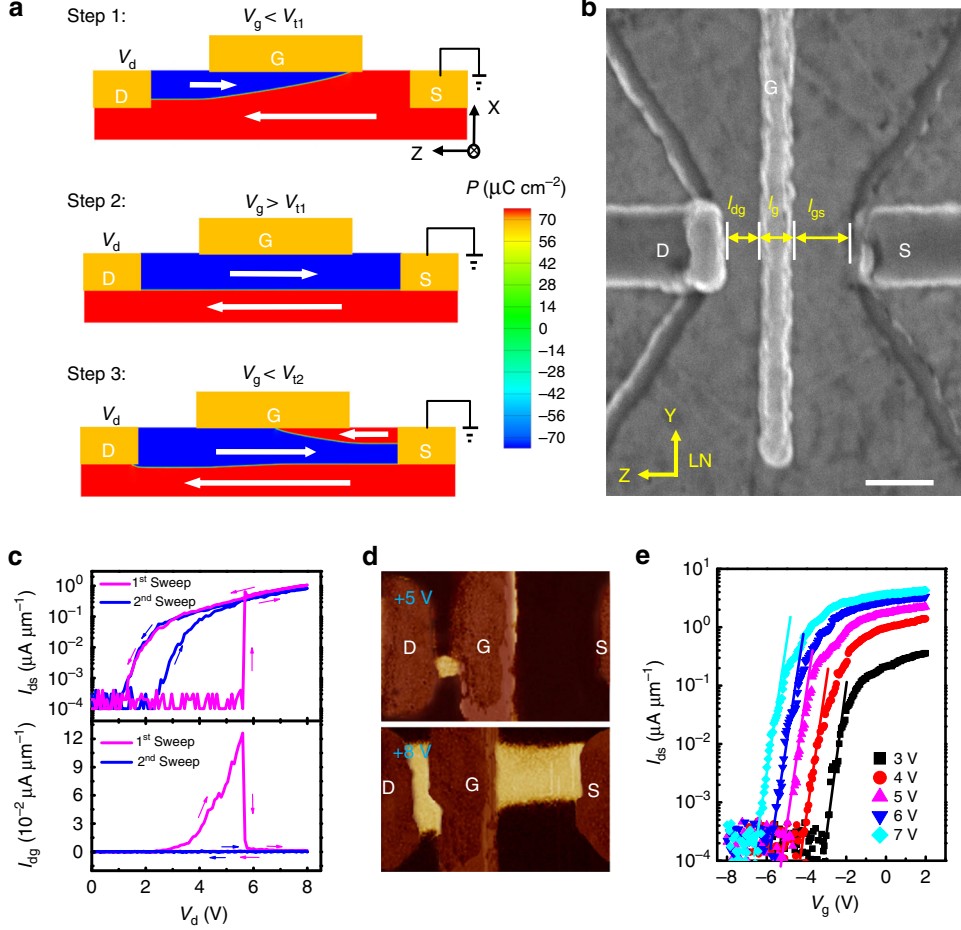

**Fig. 1 Working principles of domain-wall transistors. a** Phase-field simulations of domain wall evolution for S2 as $V_g$ varies from $-8\,V \rightarrow 0\,V \rightarrow -5\,V$ in three steps with $V_g < V_{t1}$, $V_g > V_{t1}$, and $V_g < V_{t2}$, where $V_d = 8.5\,V$, $V_s = 0$, $V_{t1} = -4.70\,V$, and $V_{t2} = -4.72\,V$. Thick arrows indicate the domain orientations. **b** Planar SEM image of a typical three-terminal LN transistor (S9). The scalar bar: 200 nm. **c** $V_d$ dependences of $I_{ds}$ and $I_{dg}$ when $V_g = V_s = 0$ during the first and second sweeps that confirm the nonvolatile D–G and D–S walls in S9 created through Steps 1 and 2. Thin arrows indicate the voltage sweeping directions. **d** In-plane PFM phase images of 180° domain reversal in the yellow-colored regions after application of various $V_d$ when $V_g = V_s = 0\,V$. **e** Semi-logarithmic $I_{ds}$–$V_g$ plots of field-effect currents across the unchanged D–S wall during backward $V_g$ sweeping of S8 under application of various $V_d$. Solid-line fits inferred an SS of 216 mV dec$^{-1}$.

conventional MOSFETs in pursuit of multigenerational computing with superior energy efficiency and high integration density has reached the Boltzmann limit at SS no less than 60 mV dec$^{-1}$ at room temperature[24]. This could be avoided in the DW transistors. To minimize the SS, $l_{dg}$, and $l_{gs}$ in S1–S5 were shortened so that the repetitive G–S domain switching would occur in Step 3 to turn off/on the complementary $I_g$ and $I_s$ by controlling either $V_s$ or $V_g$, where $I_s = I_{ds} + I_{gs}$ and $I_g = I_{dg} + I_{sg}$. For subscript description, $I_{sg}$, for example, indicates the current flowing from S to G, where $I_{sg} = -I_{gs}$. Figure 2a shows $V_g$-controlled complementary $I_s$ and $I_g$ characteristics for S1 at various applied $V_d$ when $V_s = 0$. During forward $V_g$ sweeping from $-6$ to $2\,V$, $I_s$ switches abruptly from off to an on-state above $V_{t1}$ that varies from $-0.93$ to $-4.54\,V$ as $V_d$ increases from 4 to 6.3 V; in contrast, $I_g$ switches in the opposite manner. During subsequent backward sweeping of $V_g$ from 2 to $-6\,V$, $I_s$ ($I_g$) turns off (on) below $V_{t2}$ ($-4.30$ to $-5.32\,V$). Simultaneously, several overshoots of negative $I_s$ peaks were observed near $V_{t2}$ (see upper panel of Fig. 2a), indicating the temporary current flow occurring from S to G during wall redirection (from Step 2 to Step 1) through an intermediate Step 3 (Fig. 1a). When $l_{gs}$ was shortened from 70 nm in S1 to 57 nm in S6, $V_{t2}$ increased up to $-1.95\,V$ and the $I_{sg}$ overshoots were too short to be detected

(Fig. 2b), thus indicating the adjustable hysteresis of both $V_{t1}$ and $V_{t2}$. This abrupt on-and-off state switching advances the collective domain switching behavior that breaks through the SS limit of conventional MOSFETs. Figure 2c, d show typical $V_{t1}-V_d$ and $V_{t2}-V_d$ plots for the samples listed in Table 1. The two types of plots reflect the domain hysteretic switching behavior that can be adjusted by changing either the geometrical sizes ($l_{dg}$, $l_{gs}$, $l_g$) or $V_d$.

Normally, domain switching initiates from reverse domain nucleation at the interface near D or S above a coercive field of $E_{c2}$. When the domain exceeds a critical size in Step 1, it can penetrate throughout the entire cell in Step 2 under a much smaller driving field[23]. If $l_g \gg h$, we approximately obtain

$$\frac{V_d}{l_{dg} + l_{gs}} + \frac{V_g}{l_g + l_{gs}} = E_{c1} \tag{1}$$

$$\frac{V_d}{l_{dg} + l_{gs}} + \frac{V_g}{l_{gs}} = -E_{c2} \tag{2}$$

where $E_{c1}$ is the average coercive field for D–S domain growth ($0 < E_{c1} < E_{c2}$). The solid lines in Fig. 3c, d are the best

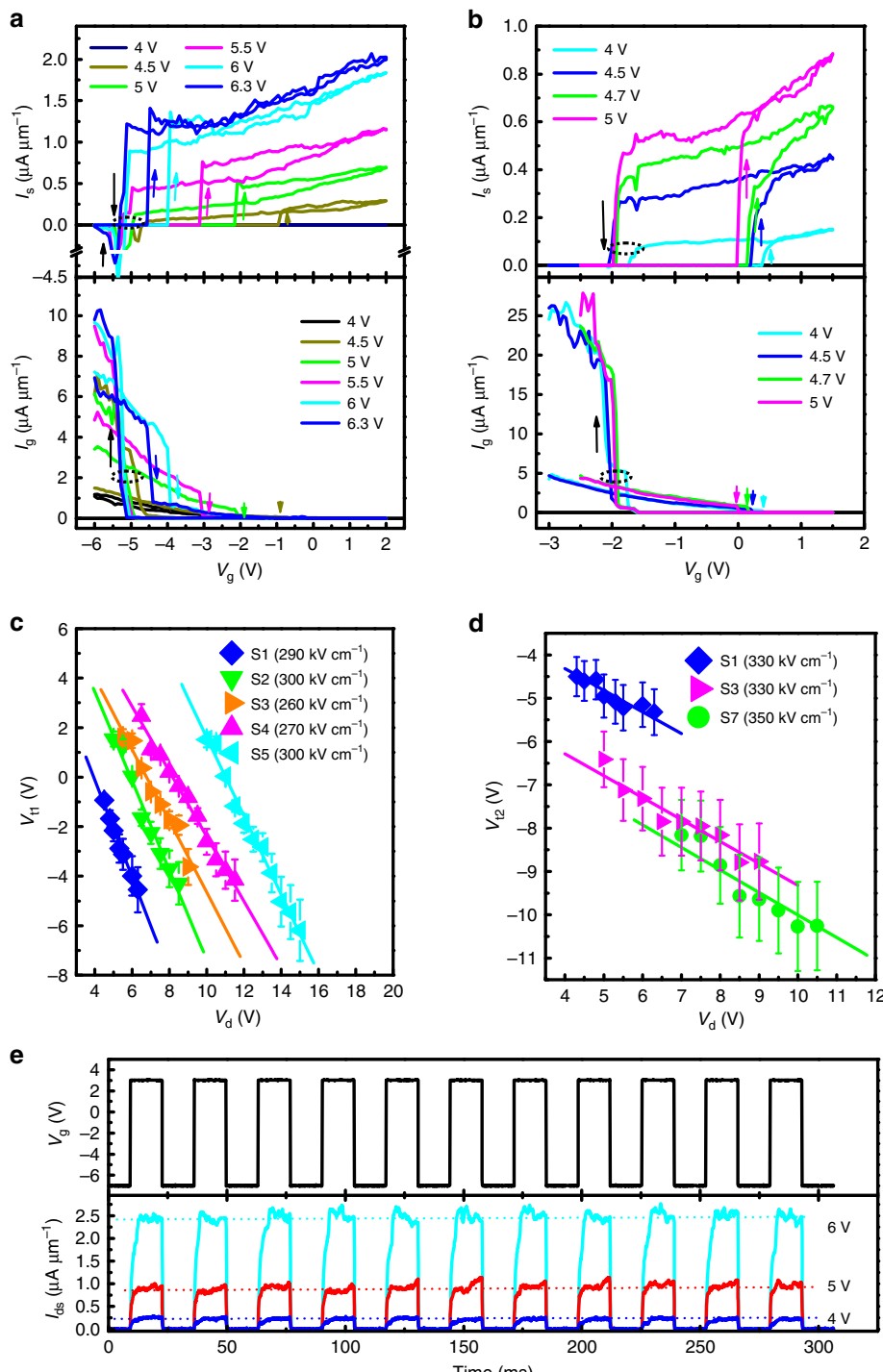

**Fig. 2 $V_g$-controlled field-effect transistors. a, b** $V_g$ dependences of complementary $I_s$ and $I_g$ under application of various $V_d$ with $V_s = 0$ for S1 and S6, respectively, where $I_s$ turns on above $V_{t1}$ (upward arrows) but turns off below $V_{t2}$ (downward arrows); this contrasts with $I_g$, which responds in the opposite manner. **c, d** $V_d$ dependences of $V_{t1}$ and $V_{t2}$ for various samples fitted using solid lines based on Eqs. (1) and (2) with the extracted coercive fields shown in parentheses. **e** Time dependence of on/off $I_{ds}$ transient for S4 when toggled using $V_g$ pulses between −7 and 3 V in full duty at a frequency of 37 Hz under application of various $V_d$ with $V_s = 0$ V. The error bars are defined as standard error of the mean.

fits to the data according to Eqs (1) and (2), from which we found that $E_{c1} = 260–300 \, kV \, cm^{-1}$ and $E_{c2} = 330–350 \, kV \, cm^{-1}$, respectively.

The semi-logarithmic $I_{ds}–V_g$ plots for S2 when $V_d = 8.5 \, V$ show the large on/off ratio (~$10^4$) and abruptness (SS = 0) of the on-to-off and off-to-on state switching over 100 cycles (Supplementary Fig. 3a). However, $V_{t1}$ and $V_{t2}$ show broad dispersions

(~0.8 V) over the cycles. These dispersions are believed to be correlated with etching damage to the D and S electrode contacts to an LN cell during device fabrication that also sets the upper limit for large current flows. To mitigate this damage, thick electroplated Ni electrodes were fabricated above Pt (Supplementary Fig. 3b). Subsequently, double $I_{ds}–V_d$ measurements showed a wall current as high as ~110 $\mu A \, \mu m^{-1}$ at 656 $kV \, cm^{-1}$

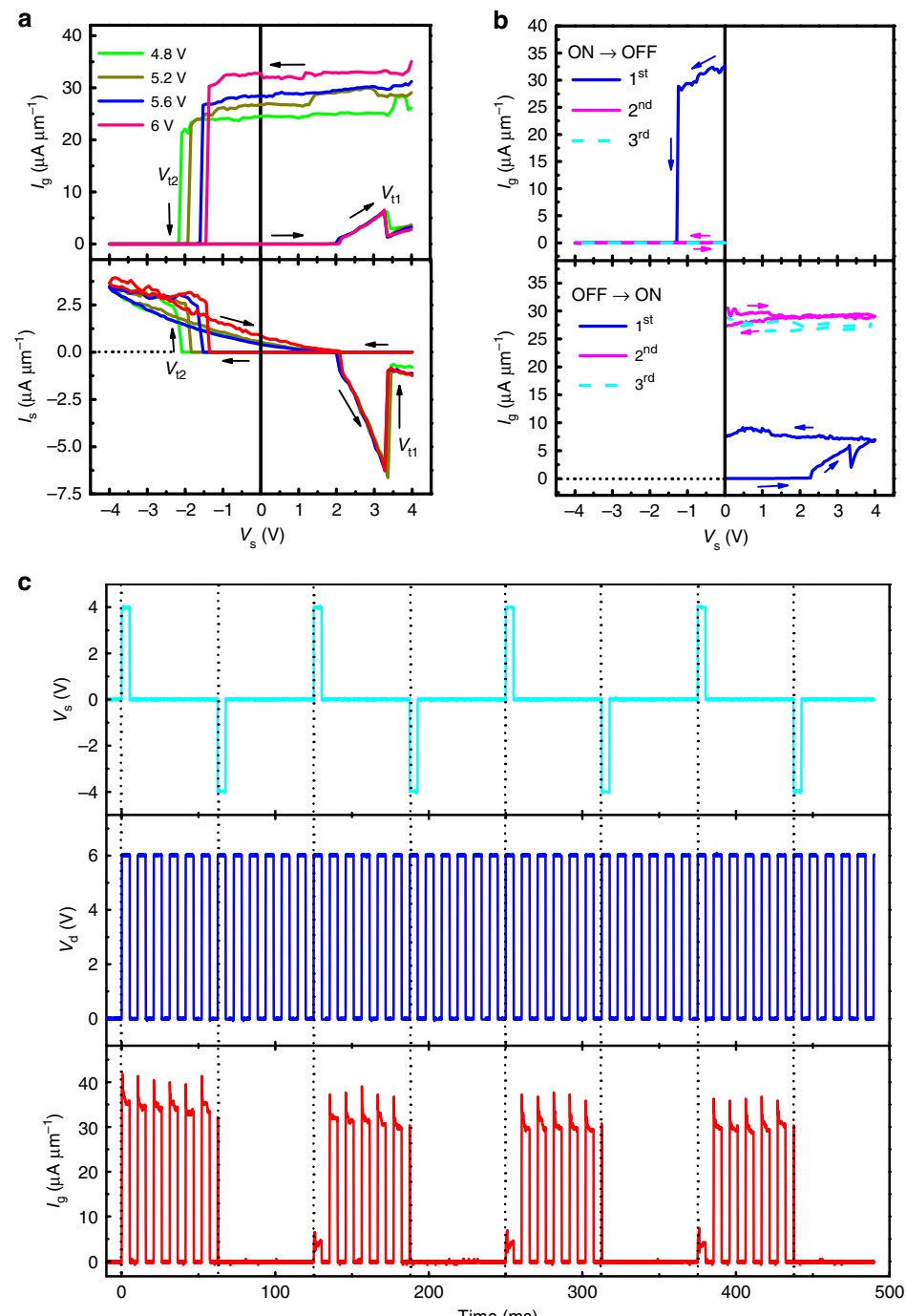

**Fig. 3 $V_s$-controlled field-effect transistors. a** $V_s$ dependences of complementary $I_g$ and $I_s$ for S10 under application of various $V_d$ with $V_g = 0$, where $I_g$ turns off below $V_{t2}$ but turns on above $V_{t1}$; this contrasts with $I_s$, which responds in the opposite manner. Arrows indicate the voltage sweeping directions. **b** On-to-off and off-to-on $I_g$ toggled using the first sweep of $V_s$, which varies either from $0 \rightarrow -4$ V $\rightarrow 0$ (upper panel) or from $0 \rightarrow 4$ V $\rightarrow 0$ (lower panel) when $V_d = 6$ V and $V_g = 0$. Each transition is nonvolatile, as confirmed by the next two repeated cycles. **c** Nonvolatile on/off $I_g$ when intermittently accessed at a repeat frequency of 96 Hz with $V_d = 6$ V and $V_g = 0$ V, as toggled using $V_s$ pulses between −4 and 4 V in 8.3% duty at a repeat frequency of 8.2 Hz.

(Supplementary Fig. 3c), and after the initial three $I_{ds}-V_d$ sweeps, the coercive field levelled off at 284 kV cm$^{-1}$ with negligible dispersion over next 120 cycles (Supplementary Fig. 3d). Unfortunately, this Ni electroplating technique cannot define fine sub-100 nm nanodevice patterns using academic facilities. Figure 2e shows successful on-and-off switching of $I_{ds}$ transients for S4 when toggled using $V_g$ pulses between −7 and 3 V at a frequency of 37 Hz under various applied $V_d$.

**$V_s$-controlled nonvolatile transistor**. In addition, the complementary on/off $I_g$ and $I_s$ can be controlled using $V_s$ between −4 and +4 V at various applied $V_d$ when $V_g = 0$, as shown in Fig. 3a for S10, where the initially-on (-off) $I_g$ ($I_s$) turns off (on) abruptly below $V_{t2}$ (<0) but back on (off) again above $V_{t1}$ (>0). There are large misfits at the start and end points in each hysteretic loop. $I_g = -I_s$ when $V_s$ increases from 2 to 3.5 V, thus providing direct evidence of D–S wall contraction from Step 2

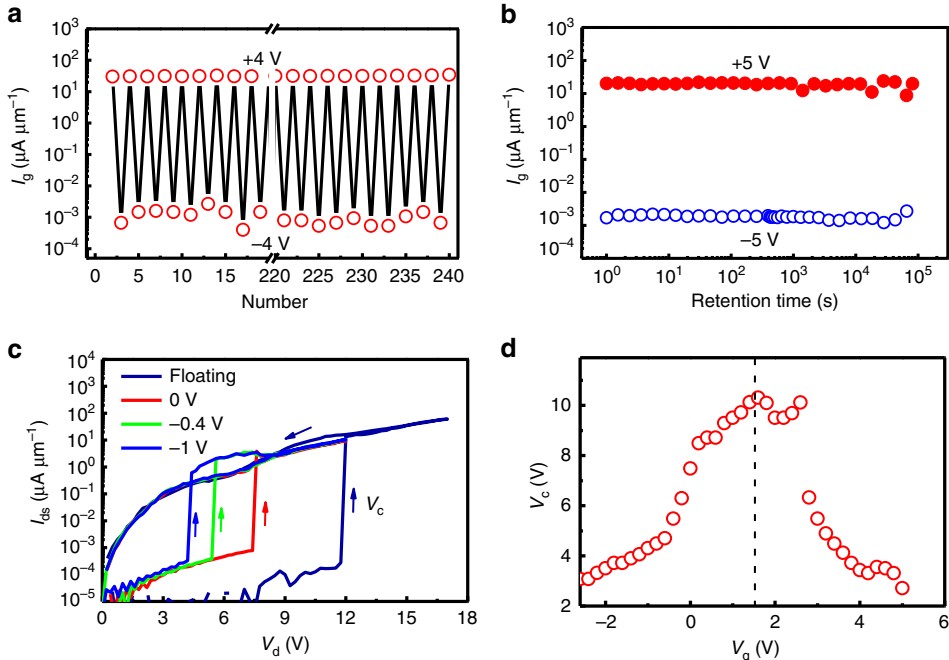

**Fig. 4 Reliability testing. a** Switching number dependence of nonvolatile on/off $I_g$ currents with $V_d = 6$ V and $V_g = V_s = 0$ when toggled using intermittent $V_s$ pulses of ±4 V. **b** Retention time dependence of nonvolatile on/off $I_g$ with $V_d = 6$ V and $V_g = V_s = 0$ when toggled using $V_s$ pulses of +5 V and −5 V. **c** Drain voltage dependence of drain-source current under various gate voltages for S13 as $V_s = 0$ V. The arrows indicate voltage sweeping directions. **d** $V_g$ dependence of $V_c$ in **c**.

into Step 1 through the intermediate Step 3 (Fig. 1a). Both on-to-off and off-to-on $I_g$ ($I_s$) switching can be realized during the first $V_s$ sweeps from $0 \rightarrow −4$ V $\rightarrow 0$ and from $0 \rightarrow 4$ V $\rightarrow 0$ in Fig. 2b (Supplementary Fig. 4a), respectively; both the on and off states are nonvolatile, as confirmed by the next two repeated cycles. The on-state current increases by more than four times after the first cycle, which appears to correlate with the wall reconstruction after removal of all applied voltages (Supplementary Fig. 4b). Figure 3c shows the nonvolatile on- and off-states when toggled using $V_s$ pulses between −4 and 4 V at a repeat frequency of 8.2 Hz, where $I_g$ was accessed intermittently from an oscilloscope at $V_d = 6$ V at a repeat frequency of 96 Hz. The switching remains stable versus switching number of up to 240 with an on/off current ratio > $10^4$, as shown in Fig. 4a; and the on- and off-states when toggled using $V_s$ pulses between −5 and 5 V are stable over retention time of 24 h, as shown in Fig. 4b. During reliability testing, a large body of data from two-terminal LN cells were adopted either at room temperature (Supplementary Fig. 5a–d) or at 85 °C (Supplementary Fig. 6a–d), where G electrodes were omitted for the convenience of electrical characterization and device fabrication. It is believed that the reliability data adopted either in three-terminal transistors or in two-terminal nanodevices are comparative due to the universality of their operation principles on the basis of domain nucleation and growth. The testing results are summarized here: (1) on/off currents in the ratio > $10^4$ are both stable over retention time of >$10^6$ s at 20 °C or >$10^5$ s at 85 °C; (2) fatigue cycles can highly reach the number of $10^{10}$ under the inhibited space-charge injection; and (3) operation speeds can be fastened from 330 ns at 500 kV cm$^{-1}$ to <5 ns at 600 kV cm$^{-1}$. Meantime, the diode-like $I_{ds}$ current in this study can suppress sneak current paths through the persistent DWs (crosstalk) when using crossbar connection of high-density LN cells (Supplementary Figs 7a–e and 8a, b).

The DW transistor can enable the non-volatile information storage among D–G, G–S, and D–S domains. Though the neutralized D–S wall is more energetically stable than the charged

D–G and G–S walls, the write voltage ($V_{ds} > V_c$) is highest. However, once the G electrode is grounded ($V_g = V_s = 0$), the local domain nucleating field between D and G strengthens under a constant $V_d$; once the head-to-head needle-like domain is nucleated above a critical size between D and G, it can grow throughout the low field G–S region due to the driving force of a depolarization field during the compensation of the increased domain wall energy at the expense of the depolarization energy[23]. In this way, $V_c$ is reduced. Figure 4c shows $I_{ds}$–$V_d$ curves for S13 at various $V_g$, where $V_c$ decreases nearly by one third as $V_g = 0$ and reduces further to 4.3 V as $V_g = −1$ V with the energy consumption of ~0.15 pJ bit$^{-1}$ (~$2PhwV_c$). The $V_c$ can be scaled down almost linearly with $l_{dg} + l_g + l_{gs}$, and the addressed cell ($V_g = −1$ V) has a much smaller write voltage than other undressed cells (floating), which can increase the reliability of the crossbar architecture (disturbance immunity during the write). Likewise, $V_c$ can also be reduced through the $E$ strengthening between G and S as $V_g > 1.5$ V, as indicated from the $V_g$–$V_c$ plot in Fig. 4d, since the tail-to-tail needle-like domain once nucleated between G and S can also grow throughout a low field D–G region in the same way.

Remarkably, the nonvolatility in the DW transistors not only reduces gate/source leakage, thus reducing standby power, but also enables programmable nanoprocessor development with computing, memory and addressing capabilities at much faster operating speeds than other charge-trapped FETs[22].

**Domain wall structure**. The nonvolatile D–G and D–S domain walls in S11 and S12 after voltage poling can be imaged by low-angle annular dark-field scanning transmission electron microscopy (LAADF-STEM), as shown in Fig. 5a–e. Before focused ion beam thinning of the two samples, the D–G and D–S domains were also inspected by in-plane PFM amplitude and phase imaging (Supplementary Fig. 9a, b). LAADF-STEM data were later collected along the [$\bar{1}$100] direction (the R3c space group in

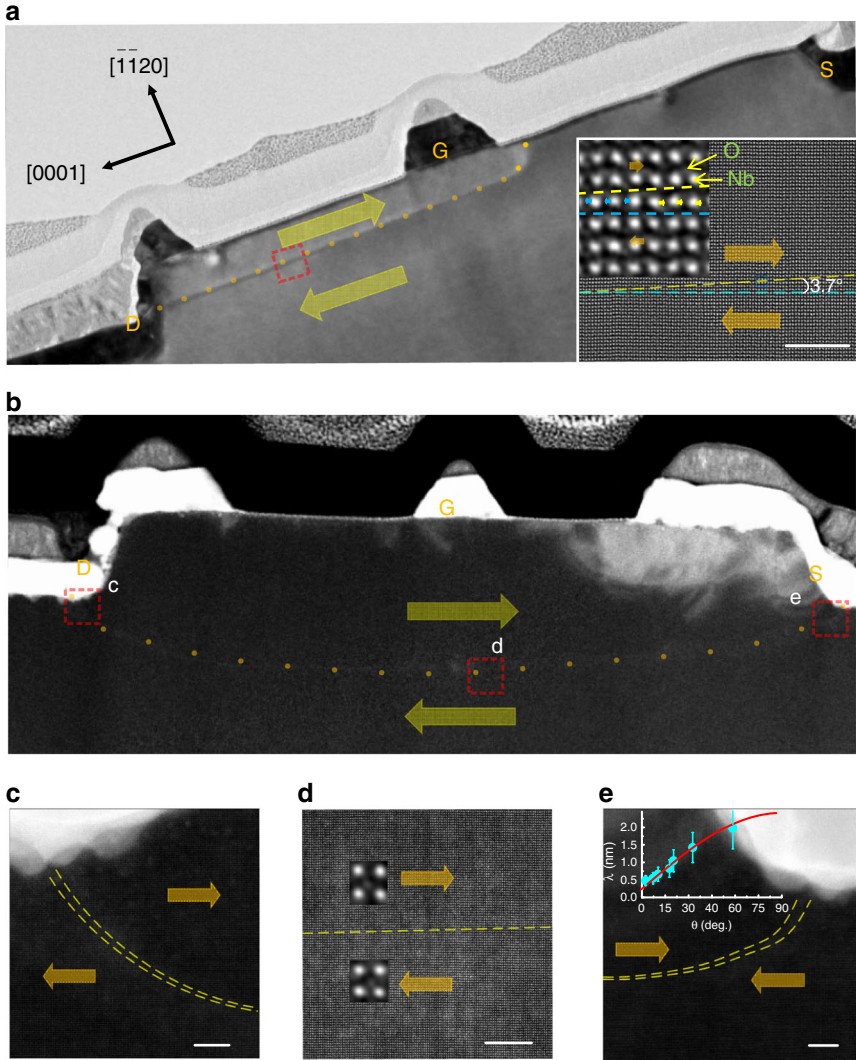

**Fig. 5 TEM characterization of the domain structure. a** Low magnification TEM image of thinned S11 lamella viewed along the $\bar{1}100$ direction after poling at $V_d = 10\ V$ and $V_g = 0\ V$ with S floating, where the dotted and dashed lines delineate a straight D-G wall running along the $000\bar{1}$ direction with a tilt angle of 3.7°, as estimated from high-resolution LAADF images of the framed region shown in the insets. **b** Decurved D-S wall in S12 after poling at $V_d = 10\ V$ and $V_g = V_s = 0\ V$. **c, d, e** High-resolution LAADF images of the framed regions in **b**, which demonstrate the presences of a charged tail-to-tail wall near D, a neutral wall below G, and a charged head-to-head wall near S; the insets show magnified dipole displacements near the wall regions in panel d and the tilt angle dependence of the wall thickness in panel e. Scale bars: 5 nm. Thick arrows indicate the domain/dipole orientations. The error bars are defined as standard deviation.

hexagonal indexes[25]) after optimization of the dose rate, the mechanical instability, and the radiation damage. The heavy Nb atoms appear brighter than the lightweight O atoms, and ferroelectric displacements of Nb cations are inverted horizontally across the domain wall (the dotted line), as indicated by the inset images in Fig. 5a. A straight D–G wall runs slightly beyond G at the mid-height of an etched cell upon the local 180° domain reversal (thick arrows), as indicated by the dotted line in Fig. 5a. Subsequent high-resolution images of the framed region in the insets show the head-to-head wall at an incline angle ($\theta$) of 3.7°, which is much higher than the typical ~1° observed in bulk crystals[11]. This increased inclination angle enhances $I_g$ in Figs. 2 and 3 by more than four orders of magnitude when compared with the $1.2 \times 10^{-5}$–$1.0 \times 10^{-2}\ \mu A\ \mu m^{-1}$ observed in bulk LN crystals[6–11]. When the domain grows throughout the entire cell, a decurved D–S wall then forms at the bottom (Fig. 5b). Subsequent high-resolution images of the framed regions in Fig. 5c–e unambiguously show a charged tail-to-tail wall near D, a neutral wall below G, and a charged head-to-head wall near S. The

LAADF-STEM image was fitted with a parametric model in which the column Nb position was derived from the intensity distribution of each atom described as a Gaussian function (Supplementary Figs 10a–d and 11a–c)[12,26], where off-center displacements of the Nb columns near the ferroelectric domain wall with the thickness $\lambda$ were analyzed using a hyperbolic tangent (tanh) function[27], as lineated between two dashed lines in Fig. 5c, e. These wall regions are rich with antiparallel dipoles for the D–G wall with a smaller inclined angle in the middle (see the inset in Fig. 5a) but with walls meandering back and forth for the decurved D–S wall with larger inclined angles near the D and S edges (Supplementary Fig. 10b)[12]. $\lambda$ broadens with increasing tilt angle, as illustrated by the plot in the inset of Fig. 5e. These plots can be fitted using a solid line based on the following equation:

$$\lambda = \lambda_0 + \alpha \cdot (2P \sin \theta), \tag{3}$$

where $\lambda_0$ is the neutral wall thickness, $\alpha$ is a coefficient, and $2P \sin \theta$ is the projected polarization charge along the wall. From the fitting, we found that $\lambda_0 = 0.25$ nm and $\alpha = 1.5 \times 10^3\ \mu m^3\ \mu C^{-1}$, implying

a single unit-cell thickness for a neutral wall ($\theta = 0°$) that thickens up to 10 unit cells when $\theta = 90°$.

## Discussion

In summary, nonvolatile LiNbO$_3$ transistors were fabricated with high complementary on/off source and gate currents that were controlled using either gate or source voltages without sub-threshold swings. The working principle of these transistors uses repetitive redirection of the conducting domain walls between D, G, and S upon ultrafast reversal of a collective ferroelectric order parameter. The observed ultrathin wall with high conductivity that could be increased further through wall angle tilt advances the voltage and size scalabilities of all-ferroelectric transistors beyond the limits of traditional MOSFETs. Each cell can perform independent data computation, storage and sensing with superior energy efficiency, stimulating processing-in-memory implementation of neuromorphic computing, general-purpose mem-computing and cybersecurity[1]. However, the digital switches are still premature at present stage when competing to the Si-based transistors: p-MOSFET is lacking for complementary operation in cascaded logic; operation voltages are very large; the hysteresis-free $I_{ds}$–$V_{gs}$ transfer characteristics are limited in some specific $V_{ds}$ as SS = 0; and the nanosecond switching speed between off- and on-states is still insufficient. Some significant improvements are required in logic for the development of next-generation domain wall nanoelectronics technologies.

## Methods

**Nanodevice fabrication**. Congruent LN single crystals containing 48.5 mol% Li$_2$O with 5 mol% MgO dopant were grown using the Czochralski technique with high-purity (99.99%) Li$_2$CO$_3$, MgO, and Nb$_2$O$_5$ powders that were melted at 1250 °C. The crystal was poled into a single domain pattern at 1180–1200 °C at a current density of 8–10 mA cm$^{-2}$ for 30 min. 200-nm-thick poly(methyl methacrylate) photoresist layers were spin-coated onto the surfaces of X-cut LN single crystals; the LN cell sizes were then defined using electron-beam lithography (EBL; JEOL 6300FS). 30-nm-thick Cr mask layers were subsequently deposited by thermal evaporation (NANO 36, Kurt J. Lesker), while any redundant Cr or photoresist layers that lay outside the written area were removed using a lift-off technique. The LN top layer that lay outside the area protected by the Cr mask layer was etched away to depths of 55–70 nm via ion milling using a reactive ion etching system (RIE-10NR, Samco, Japan). Finally, 30-nm-thick Pt top electrode layers were grown by magnetron sputtering (PVD-75, Kurt J. Lesker) at 400 °C; these layers were then etched into the D, G, and S electrodes to contact the LN mesas by repeating the EBL patterning and ion milling processes described above. To mitigate etching damage to the D and S contacts, additional 150-nm-thick electroplated Ni electrodes with widths of ~300 nm were fabricated above the Pt to strengthen the electrode contacts. All transistors were checked via planar-view SEM images (Sigma HD, Zeiss).

**Phase-field simulations**. The spatiotemporal evolution of the spontaneous polarization $P_i$ was determined using the time-dependent Ginzburg-Landau (TDGL) equations[28]:

$$\frac{\partial P_i(\mathbf{r}, t)}{\partial t} = -L \frac{\delta F}{\delta P_i(\mathbf{r}, t)} \, (i = 1, 2, 3), \tag{4}$$

where $L$ is the kinetic coefficient related to the domain mobility, $\frac{\delta F}{\delta P_i(\mathbf{r},t)}$ is the thermodynamic driving force for domain evolution, $F$ is the total free energy, $\mathbf{r}(x_1, x_2, x_3)$ is the spatial vector, and $t$ is time. The total free energy is given by the following equation:

$$F = \int (f_{Land} + f_{grad} + f_{elec}) dV, \tag{5}$$

where $f_{Land}$, $f_{grad}$, and $f_{elec}$ are the Landau energy, gradient energy, and electrical energy densities, respectively. The Landau energy density can be expressed as $f_{Land} = \alpha_1 P_3^2 + \alpha_{11} P_3^4 + \alpha_2 P_1^2$ (refs. [29,30]), where $\alpha_1$, $\alpha_{11}$, and $\alpha_2$ are the dielectric stiffnesses and the higher-order dielectric stiffnesses, respectively. The gradient energy density is related to the domain wall energy such that $f_{grad} = \frac{1}{2} G_{ijkl} P_{i,j} P_{k,l}$, where $P_{i,j}$ is the spatial derivative of the $i$th component of polarization vector $P_i$ with respect to the $j$th coordinate and $G_{ijkl}$ represent the gradient energy coefficients. The electrostatic energy can be expressed as $f_{elec} = -\frac{1}{2} \kappa_c E_i E_i - E_i P_i$, where $\kappa_c$ is the dielectric permittivity of the background materials. The electric field is calculated from $E_i = -\frac{\partial \varphi}{\partial x_i}$, where $\varphi$ is the electrical potential. It was obtained by solving the Poisson equation of

$\nabla^2 \varphi = 0$ using the specified electric potentials at the D, G, and S electrodes of the LN transistor. An open-circuit boundary condition was imposed on the top free surface area without the electrode. The elastic energy term is neglected in Eq. (5) based on the assumption that the elastic energy makes only a small contribution to the final domain wall configuration. To solve the governing equations above in the real space, a nonlinear multi-field coupling finite-element method was used. All the material constants used in the simulations are $\alpha_1 = -1.0 \times 10^9$ C$^{-2}$ m$^2$ N, $\alpha_2 = 0.9725 \times 10^9$ C$^{-2}$ m$^2$ N, $\alpha_{11} = 0.9025 \times 10^9$ C$^{-4}$ m$^6$ N, $G_{11} = G_{44} = 0.4 \times |\alpha_1|$, and $\kappa_c = 10 \times \kappa_0 = 10 \times 8.85 \times 10^{-12}$ F m$^{-1}$ (refs. [25,28]). For convenience of the calculation, the material parameters were normalized[31]. The simulations were performed using a two-dimensional system, and four-node elements with a size of 1 nm$^2$ were used to model the LN structure. To model the semi-infinite LN single crystal substrate, the charge densities on the right and left edges of the LN substrate were set to be $+P$ and $-P$, respectively. Partially reversed polarizations are assumed as the initial conditions to initiate the domain evolution process in the LN structure under application of various values of $V_d$, $V_g$, and $V_s$. The final equilibrium domain structures were obtained under the applied voltages as shown in Fig. 1a (Supplementary Fig. 1a–c).

**PFM and LAADF-STEM characterization**. The domain patterns obtained after poling using positive and negative voltages were inspected via in-plane PFM amplitude and phase imaging (Icon, Bruker) using a contact PtIr-coated silicon tip with radius of ~20 nm, a force constant of 2.8 N m$^{-1}$, and an AC amplitude of 0.5 V at 210 kHz. The LN surface was later coated with a protective Pt layer (~35 nm thick). A cross-sectional specimen was cut and thinned using a dual-beam focused ion beam/scanning microscopy (Helios G4 UX, Thermo Fisher Scientific, USA) system with Ga-ion acceleration voltages ranging from 2 to 30 kV followed by ion-milling (Gatan 691, Gatan, USA) at voltages ranging from 1.5 to 0.5 kV to remove the damaged layer. A 200 kV JEM-2100F (JEOL Ltd., Japan) microscope was used to perform the LAADF-STEM observations.

**Electrical characterization**. All current–voltage curves were measured using an Agilent B1500A semiconductor analyzer operating in voltage-sweep mode. The sweep times were 1 s when using a current amplification range of 1 µA with instrumental off-current resolutions of 20 pA. For domain switching testing, two square test pulses with rise times of 2 ns were supplied to D and G (S) using a two-channel Agilent 81110A pulse generator. The steady-state on-and-off $I_s$ ($I_g$) current transient behavior with time was observed directly using a four-channel oscilloscope (LeCroy HDO6054, USA) in series with the S (G) with 12-bit voltage resolution and a 1 GHz bandwidth. During the nanosecond-scale domain-switching period, the internal resistance of the oscilloscope in series with the sample was adjusted to 50 Ω to realize a short circuit RC time constant, and was later adjusted to 1 MΩ (100 kΩ) to enable read-out of on- and off-currents with 0.2 nA (1 nA) resolution limits as shown in Figs. 2e and 3c, respectively.

## Data availability
The authors declare that all data supporting the findings of this study are available within the paper and its supplementary information files.

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

## Acknowledgements

We thank David MacDonald, MSc, from Liwen Bianji, Edanz Editing China (www.liwenbianji.cn/ac), for editing the English text of a draft of this manuscript. This work was supported by the Basic Research Project of Shanghai Science and Technology Innovation Action (grant number 17JC1400300), the National Key R&D Program of China (grant number 2019YFA0308500), the Beijing Natural Science Foundation (grant number Z190010), the National Natural Science Foundation of China (grant numbers 61674044, 51672307, and 61904034), and the Program of Shanghai Subject Chief Scientist (grant number 17XD1400800).

## Author contributions

A.Q.J. conceived the idea for the work, performed the electrical characterization and wrote the manuscript, and, along with D.W.Z., directed the study. X.J.C. and J.J. performed the nanodevice fabrication and characterization, Q.H.Z., F.Q.M., and L.G. performed the FIB and TEM observations, and X.H. performed phase-field simulations under the direction of J.W. We thank Dr. Yan Zhang for carrying out reliability testing of the samples. X.J.C. and J.J. contributed equally to this work. All the authors discussed the results.

## Competing interests

The authors declare no competing interests.
