## [Peer Review File · Nature Communications]

Reviewers' Comments:

Reviewer #1:

Remarks to the Author:

This manuscript describes the study of transistors produced from domain walls in Mg-doped LiNbO₃. The document reports significantly enhanced diode-like wall currents, using gating that erases/creates the conducting domain walls. The authors refer to them as junctionless transistors and suggest they have low leakage, fast operating speeds, high currents, and abrupt on/off states.

This manuscript is positioned at an interesting point. First, from a topical point of view, there is growing interest in non-traditional materials for next generation logic applications; this work is well positioned in that capacity. It builds from over a decade of work on conducting domain walls in ferroelectrics, but pushes it in a new direction with some rather impressive nanofabrication and realistic device demonstrations. This said, the manuscript also sits in a bit of an awkward position in that the physics of domain wall conduction is essentially a "done" topic at this point and the manuscript provides no real advancement of the science of domain wall conduction. Instead, the merits seem to be fully placed on the demonstration of device performance as a transistor. In that regard, to be clear, the results here are not "competitive" with current state of the art. First, the voltages of operation are very large – currently devices operate at 500-600 mV range – while this work is well into the many V range. Second, the subthreshold swing of current devices bottoms out at 60 mV/dec – the authors report 216 mV/dec in one part and then (in passing) suggest 0 mV/dec later without really making a strong statement or point about this. In this regard, the question as to what this paper really accomplishes could be brought up? For this reviewer, despite these limitations, it is likely meritorious for publication in some form. This said, the authors would be well served not to mislead the reader as to the performance because of the advanced nanofabrication and the extensive data it reports – which to the best of my knowledge seems of high quality. Direct and honest comparison and acknowledgement of the limitations would be welcomed – this reviewer does not expect the authors to "solve" the entire challenge in one paper. Perhaps more helpful, in fact, would be recommendations on what could be done to improve the performance of these devices (perhaps it is just scaling things down, but these devices are already fairly small). What are the potential limits of what can be accomplished using domain wall structures as the active device? At the same time, perhaps some more scientific understanding of the observations is warranted. Right now, the manuscript comes across a bit "device-heavy" in that it just reports the observations by doesn't really show a level of control of the materials physics to achieve a specific goal.

In addition to these points, there are a few additional questions to be considered:

- Is Figure 1a really a phase field simulation? It looks like a powerpoint drawing.
- What is the thickness of the LiNbO₃ crystal? How thick is the "blue" domain or how far away from the gate is the D-S wall in Figure 1? How would varying this thickness change the device performance?
- Why was LAADF-STEM done and not simple diffraction contrast? The latter – when done with the right diffraction condition – would have clearly shown contrast between the oppositely poled domains. Instead, the data relies on the extraction of local polarization from the unit cell distortion which cannot be seen in the figure in b, c, and e.
- What is meant by the statement "These stripe regions are rich with anti-dipoles..." What are the stripe regions – the domain wall width? What do the authors mean by anti-dipoles?
- What is the point of Figure 4? The data is provided, first, to show the true nature of the domain structure – which is fine and acceptable. But beyond this, the analysis of this figure seems to be lacking. Is there something learned from this that could be used to improve the performance of the material? What does the next generation of device have to look like to be more useful?

Reviewer #2:

Remarks to the Author:

Domain wall electronics is a hot research direction, providing a possible solution to next-generation nanoelectronics. The authors follow this trend to demonstrate a domain device architecture based on ferroelectric LiNbO₃ crystals. A three terminal similar to traditional field effect transistor is used to control the generation of domain wall. The devices they showed possess two critical characteristics: 1. V_g controlled transistor without subthreshold swing, 2. V_s controlled non-volatile transistor. An extensive electrical characterization was implemented with the theoretical support (phase field simulation) and microscopic evidence (TEM). I do believe this study represent an important step for the realization of domain wall nanoelectronics. However, before I recommend the publication of this manuscript. Here are some issues to be addressed:

1. for the logic operation: it is very important to address the switching speed especially compared with current CMOS technology.
2. for low-power operation: how low energy will be used to control a bit should be calculated and the authors should also address how far they are.
3. for memory applications: ferroelectric fatigue and retention should be measured in such a configuration since the key part of device relies on a ferroelectric head-to-head domain configuration, which is not a stable state.

Reviewer #3:

Remarks to the Author:

A key feature of ferroelectric (FE) domain wall is their electrical conductivity. Electric bias not only generates stable FE domains with strongly oriented domain walls, but also can induce reversible transition between the conducting and insulating states of the domain walls as a result of electrically induced wall bending near the sample surface.

The authors here exploit that feature to create a terminal device which acts like a single pole double throw switch with reasonable repeatability.

Unfortunately, the manuscript makes a poor attempt to present this device as a) logic transistor and b) 3-terminal non-volatile memory. And fails in both aspects.

I recommend the authors to focus on only one application of the proposed device and perform a more detailed study of the domain wall device from that perspective.

Logic transistor: To qualify as a transistor and to perform as a useful compute device, the device needs to show the following:

- a) hysteresis free (or negligible) I_{ds}-V_{gs} transfer characteristics for any given V_{ds}. The threshold voltage should not vary as a function of V_{ds} (like the device presented here). Repeated cycling should not shift the I_d-V_g curves.
- b) Saturation in the I_{ds}-V_{ds} output characteristics for a given V_{gs}: There should be clear transition from low V_{ds} to high V_{ds}, and the device should transition from low impedance (at low V_{ds}) to a high impedance state (at high V_{ds}). Unfortunately, this device does not conduct at low V_{ds}, which will slow down circuits.
- c) Speed of switching between off and on-states should be in the range of sub pico-second to compete with CMOS: Domain wall velocity in MgO doped LN is typically in the range of 0.015 to 5 μm/s with the applied electric field ranging from 6 to 40 kV/cm. this is too slow for logic applications.
- d) Complementary operation in terms of a pull-up device and a pull down device is very important to build an inverter - a basic building block for logic. The proposed domain wall device does not address such functionality.
- e) Cascaded logic: one should be able to connect the inverter gate in series with the succeeding gate, and signal restoration needs to be guaranteed. This device which acts as a single pole double throw switch does not meet the criterion.

Let us now discuss application of the domain wall FET for non-volatile memory application.

a) demonstrate write and read operation clearly: Fig. 3 tries to address this, but falls short. a memory device needs to be placed in an array with word-line and bit-line clearly identified first. Then write and read operation need to be demonstrated by using the write terminal for access, program and erase; and read terminal for read out of the programmed state. It's very important to demonstrate that neighboring cells do not get disturbed (or written into) when the target cell is being accessed. Thus, inhibition schemes need to be identified clearly.

b) retention time measurement need to be shown to claim non volatility: after programming pulse, a non-destructive read pulse sequence at specified time interval need to be applied to see how the stored conductance state is maintained. Such measurement was not shown. For practical applications, both room temperature and 85C data need to be shown.

c) write and read time: with programming and erase pulse of different pulse width for a given pulse amplitude, the memory window (for example conductance ratio) need to be characterized. Then pulse amplitude need to be varied for a given pulse width and the memory window quantified. Only then, we will be able to understand the read and write latency as a function of the write energy for the proposed memory. Short pulse height and short pulse duration may not only affect memory read window, but also affect retention. That needs to be characterized.

d) variation is the most important attribute after demonstration of basic memory operation as noted above. It's not clear how the nucleation and growth of the domains is guaranteed for reproducible characteristics as a function of repeated switching. Endurance measurement is also advised to ascertain the application space of the proposed FE domain wall non-volatile memory.

In summary, a detailed effort focusing on one type of device (logic or memory) is being recommended to markedly improve the quality of the manuscript.

Reply to the referees (Nature Communications NCOMMS-19-39458):

We thank the three reviewers for their detailed and constructive comments for our manuscript (NCOMMS-19-39458). While the reviewers found good interests from our work, they also raised several concerns regarding the device physics, the reliability issue (device switching speed, retention, and endurance measurements), and the uncertainty of the focused research between memory and logic devices. We accepted these criticisms and revised the entire manuscript to clarify the device working principle (phase field simulations), the method to estimate domain wall (DW) thickness, reliability for retention, domain switching time, endurance, and crosstalk. With these revisions, one could find that this study is focused at new memory devices now that enable high-throughput, energy-efficient and area-efficient information processing. Finally, we also pointed out the shortcomings of this kind of new devices in comparison to the current CMOS technology for the application in logic only.

We appended detailed answers to each comment below.

Reply to the Referee (NCOMMS-19-39458)

Reviewer #1 (Remarks to the Author):

Comment from the reviewer:

This manuscript describes the study of transistors produced from domain walls in Mg-doped LiNbO₃. The document reports significantly enhanced diode-like wall currents, using gating that erases/creates the conducting domain walls. The authors refer to them as junctionless transistors and suggest they have low leakage, fast operating speeds, high currents, and abrupt on/off states.

This manuscript is positioned at an interesting point. First, from a topical point of view, there is growing interest in non-traditional materials for next generation logic applications; this work is well positioned in that capacity. It builds from over a decade of work on conducting domain walls in ferroelectrics, but pushes it in a new direction with some rather impressive nanofabrication and realistic device demonstrations. This said, the manuscript also sits in a bit of an awkward position in that the physics of domain wall conduction is essentially a “done” topic at this point and the manuscript provides no real advancement of the science of domain wall conduction. Instead, the merits seem to be fully placed on the demonstration of device performance as a transistor. In that regard, to be clear, the results here are not “competitive” with current state of the art. First, the voltages of operation are very large – currently devices

operate at 500-600 mV range –while this work is well into the many V range. Second, the subthreshold swing of current devices bottoms out at 60 mV/dec – the authors report 216 mV/dec in one part and then (in passing) suggest 0 mV/dec later without really making a strong statement or point about this. In this regard, the question as to what this paper really accomplishes could be brought up? For this reviewer, despite these limitations, it is likely meritorious for publication in some form. This said, the authors would be well served not to mislead the reader as to the performance because of the advanced nanofabrication and the extensive data it reports – which to the best of my knowledge seems of high quality. Direct and honest comparison and acknowledgement of the limitations would be welcomed – this reviewer does not expect the authors to “solve” the entire challenge in one paper. Perhaps more helpful, in fact, would be recommendations on what could be done to improve the performance of these devices (perhaps it is just scaling things down, but these devices are already fairly small). What are the potential limits of what can be accomplished using domain wall structures as the active device? At the same time, perhaps some more scientific understanding of the observations is warranted. Right now, the manuscript comes across a bit “device-heavy” in that it just reports the observations by doesn’t really show a level of control of the materials physics to achieve a specific goal.

Answer from the authors:

We thank the referee to read the entire manuscript carefully and very helpful comments. We reorganized/revised the entire manuscript to improve its clarity for the better understanding of the physics (Supplementary notes 1 and 6). The major breakthrough in this study is the finding of the nonvolatile and erasable DW transistors that can enable processing-in-memory implementation in breaking Neumann bottleneck and the memory wall, just like other studies in RRAMs, MRAMs, PCRAMs, etc. (refs. 1, 21, 22). We understood the shortcomings of this new device for the application in logic in comparison to the current CMOS (see Conclusions), especially due to their large subthreshold swings and high operation voltages. But we have shown our effort to scale them down under the control of V_g and V_s , besides the device dimensions. For example, Fig. 4c, d shows the method how to reduce the coercive voltage significantly in three-terminal transistors according to the ferroelectric principle. In this sense, this work provides not only the new devices but also the new physics. Undoubtedly, the integration of all-ferroelectric transistors is an important step toward the development of next-generation domain wall nanoelectronics technologies, where p-MOSFET and n-MOSFET could be made using tail-to-tail and head-to-head DWs, intriguing the combined integration of ferroelectric sensors, actuators, electro-optic modulators, and memories independent of Si circuits in the future. We fully agree with the reviewer that these digital switches are still premature in logic at this stage, but we are quite optimistic that the performance of these devices can be improved greatly in the near future. We wish that the reviewer could find the high merit of this work.

Comment from the reviewer:

In addition to these points, there are a few additional questions to be considered:
- Is Figure 1a really a phase field simulation? It looks like a powerpoint drawing.

- What is the thickness of the LiNbO₃ crystal? How thick is the “blue” domain or how far away from the gate is the D-S wall in Figure 1? How would varying this thickness change the device performance?

Answer from the authors:

Figure 1a is indeed from a phase field simulation. Some fine structures for the domain walls and potentials can be discerned from inset figures in Supplementary note 1. The bulk LN thickness is ~0.4 mm, and the geometrical sizes of the nanodevice (S2) can be found in Table I. The D-G domain thickness is 46 nm, and the D-S domain thickness equals the etching depth (h) of the cell (Supplementary note 1). The D-G domain thickness could change with the length (l_{dg}) between the drain (D) and gate (G), but its aspect ratio is nearly constant. Therefore, the etching depth should be larger than the D-G domain thickness that depends on l_{dg} . We added these comments in Supplementary note 1:

“the head-to-head wedged domain initiating from D grows up below G with the length of 355 nm and the thickness of 46 nm, as shown in Supplementary fig. 1a (left panel). The aspect ratio for the needle-like domain growth is 7.7 that is generally constant irrespective of the geometrical sizes¹. In this sense, the smaller l_{dg} implies the thinner D-G domain, and the etching depth (h) of the device should be larger than the thickness of the D-G domain. The insets in Supplementary fig. 1a-c show the wall thicknesses of 0.72–1.06 nm when $\theta = 0-8.4^\circ$, roughly in agreement with the TEM estimations in Fig. 5e in main text.”

Comment from the reviewer:

- Why was LAADF-STEM done and not simple diffraction contrast? The latter – when done with the right diffraction condition - would have clearly shown contrast between the oppositely poled domains. Instead, the data relies on the extraction of local polarization from the unit cell distortion which cannot be seen in the figure in b, c, and e.

Answer from the authors:

Indeed, a dark-field image of the D-G domain wall in Fig. 5a clearly shows light contrast between the two oppositely poled domains for sample S11. Unfortunately, the domain contrast is blurred for sample S12, because the lamella was too thin and winding but was necessary for high-resolution LAADF imaging. From the LAADF-STEM images, we can extract Nb ion positions for the calculation of wall thicknesses accurately. The calculated results are shown in Fig. 5c-d with the method fully described in the Supplementary note 6. For this clarification, we added the sentences in lines 9-15 on page 11:

“The LAADF-STEM image was fitted with a parametric model in which the column Nb position was derived from the intensity distribution of each atom described as a Gaussian function (Supplementary figs. 10a-d and 11a-c)^{12,26}, where off-center displacement of the Nb columns near the ferroelectric domain wall with the thickness λ were analyzed using a hyperbolic tangent (\tanh) function²⁷, as lineated between two dashed lines in Fig. 5c,e.”

Comment from the reviewer:

- What is meant by the statement “These stripe regions are rich with anti-dipoles...” What are the stripe regions – the domain wall width? What do the authors mean by anti-dipoles?
- What is the point of Figure 4? The data is provided, first, to show the true nature of the domain structure – which is fine and acceptable. But beyond this, the analysis of this figure seems to be lacking. Is there something learned from this that could be used to improve the performance of the material? What does the next generation of device have to look like to be more useful?

Answer from the authors:

We are sorry for this confusion. The sentence has been corrected as “*These wall regions are rich with antiparallel dipoles for the D-G wall with a smaller inclined angle in the middle (see the inset in Fig. 5a) but with walls meandering back and forth for the decurved D-S wall with larger inclined angles near the D and S edges (Supplementary figs. 10b)¹².*” (lines 15-18 on page 11). The two dashed lines either in Fig. 5c or in Fig. 5e delineated the wall thicknesses (within the stripe region).

Figure 5a-e (previous Fig. 4a-e) is more understandable now with the input of Supplementary note 6 in this revised manuscript, where the wall meandering and thickening nature against the tilted angle were very clear (Supplementary figs. 10b-d). These features are helpful for the understanding of the origin of large wall currents in LN (refs. 10-12) that could be increased again in the future for the better performance of high power nanodevices.

Reviewer #2 (Remarks to the Author):

Comment from the reviewer:

Domain wall electronics is a hot research direction, providing a possible solution to next-generation nanoelectronics. The authors follow this trend to demonstrate a domain device architecture based on ferroelectric LiNbO₃ crystals. A three terminal similiar to traditional field effect transistor is used to control the generation of domain wall. The devices they showed possess two critical characteristics: 1. V_g controlled transistor without subthreshold swing, 2. V_s controlled non-volatile transistor. An extensive electrical characterization was implemented with the theoretical support (phase field simulation) and microscopic evidence (TEM). I do believe this study represent an important step for the

realization of domain wall nanoelectronics. However, before I recommend the publication of this manuscript. Here are some issues to be addressed:

1. for the logic operation: it is very important to address the switching speed especially compared with current CMOS technology.

Answer from the authors:

We thank the referee very much to read the entire manuscript carefully and give positive comments to our works. We showed the shortest domain switching time of < 5 ns in Supplementary note 5 (Supplementary figs. 5b and 6d) from two-terminal nanodevices where G was omitted for the convenience of electrical characterization and device fabrication. During testing, we found that the Pt electrodes were easily damaged in three-terminal devices due to huge wall currents at high switching voltages. We believed that the switching data adopted either in three-terminal transistors or in two-terminal nanodevices are comparative due to the universality of their operation principles on the basis of domain nucleation and growth. We also pointed out the limit of these devices at this stage compared with current sub-picosecond CMOS technology (line 15 on page 12), “*the nanosecond switching speed between off- and on-states is still insufficient*”, though “*These walls can be reversibly created, positioned and shaped using electric fields on a femtosecond time scale¹⁵*” in the theory (line 13-14 on page 3).

Comment from the reviewer:

2. for low-power operation: how low energy will be used to control a bit should be calculated and the authors should also address how far they are.

Answer from the authors:

From the last line on page 9, we calculated the energy consumption of ~ 0.15 pJ/bit with the formula of $2PhwV_c$, where hw is the cross-sectional area of a mesa-like cell, P is the polarization, and V_c is a minimum write voltage (coercive voltage). The value can be scaled down with the cell dimensions.

Comment from the reviewer:

3. for memory applications: ferroelectric fatigue and retention should be measured in such a configuration since the key part of device relies on a ferroelectric head-to-head domain configuration, which is not a stable state.

Answer from the authors:

In this revised manuscript, the data from fatigue and retention are given in Supplementary Note 5 (Supplementary fig. 5a, c), as summarized in the 1st paragraph on page 9:

“1) on/off currents in the ratio $> 10^4$ are both stable over retention time of $>10^6$ s at 20°C or $>10^5$ s at 85°C ; 2) fatigue cycles can highly reach the number of 10^{10} under the inhibited space-charge injection; and 3) operation speeds can be fastened from 330 ns at 500 kV/cm to <5 ns at 600 kV/cm. Meantime, the diode-like I_{ds} current in this study can suppress sneak current paths through the persistent DWs (crosstalk) when using crossbar connection of high-density LN cells (Supplementary figs. 7a-e and 8a, b).”

The DW transistor can enable the non-volatile information storage among D-G, G-S, and D-S domains. We understand that the charged D-G and D-S domains are energetically unstable than the neutralized D-S domain in theory. But the coercive voltage across the entire D-S domain is highest. Therefore, we found a way to reduce V_c from 12 V to 4.3 V for S13 across the total cell size of 280 nm ($l_{dg}+l_g+l_{gs}$) in Fig. 4c, d (see the 2nd paragraph on page 9). Either, the V_c can be scaled down linearly with the lateral size ($l_{dg}+l_g+l_{gs}$) of the device.

Reviewer #3 (Remarks to the Author):

Comment from the reviewer:

A key feature of ferroelectric (FE) domain wall is their electrical conductivity. Electric bias not only generates stable FE domains with strongly oriented domain walls, but also can induce reversible transition between the conducting and insulating states of the domain walls as a result of electrically induced wall bending near the sample surface.

The authors here exploit that feature to create a terminal device which acts like a single pole double throw switch with reasonable repeatability.

Unfortunately, the manuscript makes a poor attempt to present this device as a) logic transistor and b) 3-terminal non-volatile memory. And fails in both aspects.

I recommend the authors to focus on only one application of the proposed device and perform a more detailed study of the domain wall device from that perspective.

Answer from the authors:

We thank the referee very much for his/her careful reading of the entire manuscript and very helpful comments. We understand that the present ferroelectric digital switches cannot compete to the CMOS logic yet. We reorganized/revised the entire manuscript to focus our study at processing-in-memory implementation to break Neumann bottleneck and the memory wall, just like other RRAMs, MRAMs, PCRAMs, etc. (refs. 1, 21, 22). In avoidance of misleading of the readers, we added the following sentences in the introduction (lines 1-6 on page 3):

“The frequent data shuttling between the physically separated processing and memory units

in traditional digital computers incurs considerable penalties on the energy efficiency and data bandwidth, which is further intensified by the increasing disparity between the speed of the memory unit and the processor¹. New computing approaches require new memory devices to enable high-throughput, energy-efficient and area-efficient information processing.”

Comment from the reviewer:

Logic transistor: To qualify as a transistor and to perform as a useful compute device, the device needs to show the following:

- a) hysteresis free (or negligible) I_{ds} - V_{gs} transfer characteristics for any given V_{ds} . The threshold voltage should not vary as a function of V_{ds} (like the device presented here). Repeated cycling should not shift the I_d - V_g curves.
- b) Saturation in the I_{ds} - V_{ds} output characteristics for a given V_{gs} : There should be clear transition from low V_{ds} to high V_{ds} , and the device should transition from low impedance (at low V_{ds}) to a high impedance state (at high V_{ds}). Unfortunately, this device does not conduct at low V_{ds} , which will slow down circuits.
- c) Speed of switching between off and on-states should be in the range of sub pico-second to compete with CMOS: Domain wall velocity in MgO doped LN is typically in the range of 0.015 to 5 $\mu\text{m/s}$ with the applied electric field ranging from 6 to 40 kV/cm. this is too slow for logic applications.
- d) Complementary operation in terms of a pull-up device and a pull down device is very important to build an inverter - a basic building block for logic. The proposed domain wall device does not address such functionality.
- e) Cascaded logic: one should be able to connect the inverter gate in series with the succeeding gate, and signal restoration needs to be guaranteed. This device which acts as a single pole double throw switch does not meet the criterion.

Answer from the authors:

We agree with the referee that that the present ferroelectric transistors cannot compete to the CMOS logic yet. We make this statement clear in the conclusions (lines 11-16 on page 12):

“However, the digital switches are still premature at present stage when competing to the Si-based transistors: p-MOSFET is lacking for complementary operation in cascaded logic; operation voltages are very large; the hysteresis-free I_{ds} - V_{gs} transfer characteristics are limited in some specific V_{ds} as $SS = 0$; and the nanosecond switching speed between off- and on-states is still insufficient. Some significant improvements are required in logic for the development of next-generation domain wall nanoelectronics technologies.”

Comment from the reviewer:

Let us now discuss application of the domain wall FET for non-volatile memory application.

- a) demonstrate write and read operation clearly: Fig. 3 tries to address this, but falls short. a

memory device needs to be placed in an array with world-line and bit-line clearly identified first. Then write and read operation need to be demonstrated by using the write terminal for access, program and erase; and read terminal for read out of the programmed state. It's very important to demonstrate that neighboring cells do not get disturbed (or written into) when the target cell is being accessed. Thus, inhibition schemes need to be identified clearly.

Answer from the authors:

For the demonstration of write and read operation clearly, Figure 4a-d is added in this revised manuscript for the further discussion of the nonvolatile domain transistor with good performance. This DW transistor can enable non-volatile information storage among D-G, G-S, and D-S domains. We understand that the charged D-G and D-S domains are energetically unstable than the neutralized D-S domain in theory, and that the write voltage for the D-S domain is highest. To overcome this shortcoming, we added Fig. 4c, d in this revised manuscript for the exhibition how to reduce V_c for the D-S domain from 12 V to 4.3 V using the V_g access for S13 ($I_{dg}+I_g+I_{gs}=280$ nm) (see the 2nd paragraph on page 9). It is remarked that the V_c can be either scaled down linearly with the lateral size of the device. For the demonstration of the crosstalk immunity between neighboring cells in the crossbar scheme, we adopted a large body of the data from two-terminal nanodevices, as shown in Supplementary note 5 (Supplementary figs. 7a-e and 8a, b) where G was omitted for the convenience of electrical characterization and device fabrication. During testing, we found that the Pt electrodes were easily damaged in three-terminal devices due to huge wall currents at high switching voltages. We believed that the reliability data adopted either in three-terminal transistors or in two-terminal nanodevices are comparative due to the universality of their operation principles on the basis of domain nucleation and growth. We identified the superiority of diode-like wall currents that can inhibit the crosstalk in the arrays.

Comment from the reviewer:

b) retention time measurement need to be shown to claim non volatility: after programming pulse, a non-destructive read pulse sequence at specified time interval need to be applied to see how the stored conductance state is maintained. Such measurement was not shown. For practical applications, both room temperature and 85C data need to be shown.

Answer from the authors:

We showed the retention data either at room temperature ($>10^6$ s) or at 85°C ($>10^5$ s) (Supplementary figs. 5a and 6c) in this revised manuscript, where “on” and “off” currents are stable with the on/off ratio $>10^6$. The ratio largely depends on the device leakage invoked during etching process.

Comment from the reviewer:

c) write and read time: with programming and erase pulse of different pulse width for a given pulse amplitude, the memory window (for example conductance ratio) need to be characterized. Then pulse amplitude need to be varied for a given pulse width and the memory window quantified. Only then, we will be able to understand the read and write latency as a function of the write energy for the proposed memory. Short pulse height and short pulse duration may not only affect memory read window, but also affect retention. That needs to be characterized.

Answer from the authors:

The measurements of read and write time either at room temperature or at 85°C are given in Supplementary figs. 5b and 6d. In ferroelectrics, the switching time always obeys Merz's law², $\tau_1 = \tau_0 \exp\left(\frac{E_a}{E}\right)$, where E_a is the activation field (6.9 MV/cm), and τ_0 is the shortest time of $\sim 10^{-13}$ s (ref. 3). In this study, we confirmed the validity of this law either in LN bulk crystals or in LOI thin films (Supplementary note 5). The write energy is ~ 0.15 pJ/bit, as estimated from the formula of $2PhwV_c$ (the last line on page 9), where hw is the cross-sectional area of a mesa-like cell, P is the polarization, and V_c is a minimum write voltage (coercive voltage). According to our observations, there is no retention problem after the write, just like other FRAMs (using the same principle of the polarization reversal).

Comment from the reviewer:

d) variation is the most important attribute after demonstration of basic memory operation as noted above. It's not clear how the nucleation and growth of the domains is guaranteed for reproducible characteristics as a function of repeated switching. Endurance measurement is also advised to ascertain the application space of the proposed FE domain wall non-volatile memory.

Answer from the authors:

Endurance measurement is given in Supplementary fig. 5c with the switching cycles of up to 10^{10} in this study. But this is not the ultimate limit. We cannot continue this testing due to the electrical damage of the Pt electrodes after a long-time testing. Though LN films are very good in atomic layer roughness, most difficulties in controlling the electrical variation arise from the nanofabrication techniques using academic facilities, as discussed in Supplementary note 3. Actually, the endurance can be increased to 10^{14} in PZT (Fujitsu FRAMs) if using other oxide metal electrodes.

Comment from the reviewer:

In summary, a detailed effort focusing on one type of device (logic or memory) is being recommended to markedly improve the quality of the manuscript.

Answer from the authors:

We appreciate this comment and revise the whole manuscript accordingly. The whole work is focused at the processing-in-memory application at this stage. A large body of the data is input in Supplementary note 5 for the reliability testing. We wish that the reviewer could find the high quality of this work in our best effort.

Reviewers' Comments:

Reviewer #1:

Remarks to the Author:

I have read the rebuttal and revised text and find it satisfactory for publication.

Reviewer #2:

Remarks to the Author:

All the questions are addressed properly. I do recommend the publication of this manuscript.

Reply to the referees (NCOMMS-19- 39458A):

We thank the three reviewers very much for their careful reading of the work, and understand that they were satisfactory and recommended acceptance of the revised version manuscript.

Reply to the Referee (NCOMMS-19-39458A)

Comments from the reviewers:

Reviewer #1 (Remarks to the Author):

I have read the rebuttal and revised text and find it satisfactory for publication.

Reviewer #2 (Remarks to the Author):

All the questions are addressed properly. I do recommend the publication of this manuscript.

Reviewer #3 (Remarks to the Author):

Reviewer #3 does not reply so we invite other reviewer to check it. The other reviewer is satisfied with it and supports publication.

Answer from the authors:

We appreciate these comments and thank all reviewers to support for publication.